

# Ferroptosis and its implications in bone-related diseases

Zihao Wang[1,2,*], Qiupeng Yan[2,3,*], Zhen Wang[1,2], Zunguo Hu[1], Chenchen Wang[2,4], Xue Zhang[2,4], Xueshuai Gao[2,4], Xue Bai[2,4], Xiaosu Chen[2,4], Lingyun Zhang[2], Danyue Lv[5], Huancai Liu[1] and Yanchun Chen[2,4]

[1] Shandong Second Medical University, Department of Joint Surgery, Affiliated Hospital of Shandong Second Medical University, School of Clinical Medicine, Weifang, Shandong, China
[2] Shandong Second Medical University, Neurologic Disorders and Regenerative Repair Lab of Shandong Higher Education, Weifang, Shandong, China
[3] Shandong Second Medical University, Department of Teaching and Research Section of Introduction to Basic Medicine, School of Basic Medical Sciences, Weifang, Shandong, China
[4] Shandong Second Medical University, Department of Histology and Embryology, School of Basic Medical Sciences, Weifang, Shandong, China
[5] Shandong Second Medical University, Clinical Medicine, School of Clinical Medicine, Weifang, Shandong, China
* These authors contributed equally to this work.

Corresponding authors
Huancai Liu,
fyliuhuancai@sdsmu.edu.cn
Yanchun Chen,
cyc7907@sdsmu.edu.cn

## ABSTRACT

Ferroptosis, a recently recognized form of regulated cell death (RCD) characterized by iron-dependent lipid peroxide accumulation, has emerged as a noteworthy regulator in various bone-related diseases, including osteoporosis (OP), osteoarthritis (OA), and osteosarcoma (OS). OS primarily afflicts the elderly, rendering them susceptible to fractures due to increased bone fragility. OA represents the most prevalent arthritis in the world, often observed in the aging population. OS predominantly manifests during adolescence, exhibiting an aggressive nature and bearing a significantly unfavorable prognosis. In this review article, we present an overview of the characteristics and mechanism of ferroptosis and its involvement in bone-related diseases, with a particular focus on OP, OA, and OS. Furthermore, we summarize chemical compounds or biological factors that impact bone-related diseases by regulating ferroptosis. Through an in-depth exploration of ferroptosis based on current research findings, this review provides promising insights for potential therapeutic approaches to effectively manage and mitigate the impact of these bone-related pathological conditions.

## INTRODUCTION

Bone-related diseases constitute a diverse group of medical conditions that primarily target the skeletal system, often resulting in a range of physical, psychological, and economic impairments (*Wang et al., 2024*). These conditions may affect individuals across a wide range of age groups, from children to the elderly, and they vary in severity and long-term effects (*Wang et al., 2024*). This review primarily focuses on bone-related diseases such as osteoarthritis (OA), osteoporosis (OP), and osteosarcoma (OS).

OP is characterized by a reduction in bone density and structural integrity, leading to heightened bone fragility (*Black & Rosen, 2016*). Notably, OP stands as a primary risk factor for fractures, underscoring its significant clinical relevance (*Liu et al., 2022*). Globally, osteoporotic fractures of the hip, spine and forearm are associated with impaired gait, physical deformities, chronic pain and disability, loss of independence and reduced quality of life (*Black & Rosen, 2016*). OP is one of the leading causes of hip fractures, accounting for 5% of all deaths in both men and women (*GBD 2017 Disease and Injury Incidence and Prevalence Collaborators, 2018*), of which 21–30% die within a year (*Viswanathan et al., 2018*).

OA stands as the most prevalent degenerative joint disorder globally. It afflicts over 500 million individuals worldwide, manifesting in varying degrees of severity and affecting approximately 7% of the world's population (*Hunter, March & Chew, 2020*). Due to the emergence of contemporary health-related challenges, such as increasing aging population, heightened life expectancy, and mounting obesity rates, the frequency and occurrence of OA are progressively escalating (*Hunter & Bierma-Zeinstra, 2019*). As a systemic joint disease, OA affects most joint tissues, including the menisci, ligaments, cartilage, subchondral bone, and infrapatellar fat pad (*Hunter & Bierma-Zeinstra, 2019*). The etiology and pathogenesis of osteosarcoma OA are complex and not fully understood, however, it is widely recognized that cartilage degeneration plays a crucial role in this process (*Jiang, 2022*). A disequilibrium in cartilage homeostasis exacerbates the development of OA when processes that promote matrix degradation surpass the capacity of chondrocyte synthesis (*Zhang et al., 2022*).

OS is one of the most common malignant bone tumors. It exhibits a bimodal age distribution, with higher incidences observed among children, adolescents, and individuals over the age of 60, while middle-aged individuals experience a lower incidence (*Kansara et al., 2014*). The global incidence of OS is approximately 1–3 cases per million per year (*Kansara et al., 2014*). OS predominantly occurs in long bones such as the distal femur and proximal tibia, where less frequently in the skull, jaw, and pelvis (*Mutsaers & Walkley, 2014*). The precise cellular origin of the tumor remains incompletely understood (*Mutsaers & Walkley, 2014*). Existing evidence supports the theory that OS originates from mesenchymal stem/stromal cells (MSCs) or, with greater reliability, osteoblastic progenitors (*Gaebler et al., 2017*; *Mutsaers & Walkley, 2014*).

Numerous degenerative illnesses and cancers are influenced by regulated cell death (RCD), which is crucial to the growth and development of multicellular organisms. Throughout the last few decades, over 20 distinct forms of RCD have been identified (*Koren & Fuchs, 2021*), including apoptosis, necroptosis, ferroptosis, pyroptosis and autophagy-dependent cell death *etc.* which are classified based on various morphological traits, biomarkers, or regulatory process (*Chen et al., 2020*; *Li, Wei & Jiang, 2019a*). In this review, we focus on ferroptosis, a non-apoptotic process driven by iron-dependent lipid peroxidation, initially proposed as a form of RCD in 2012 (*Coradduzza et al., 2023*; *Dixon et al., 2012*). The characteristics of ferroptosis include iron accumulation and lipid peroxidation. During ferroptosis, accumulated iron ions promote lipid peroxidation. The

extensive formation of lipid peroxides, in turn, leads to cell membrane rupture and the release of pro-inflammatory signals, ultimately resulting in cell death (*Li et al., 2023*).

Research has revealed that ferroptosis plays a significant role in bone-related diseases (*Gao et al., 2022*). Dysregulation of ferroptosis-related biological processes, such as iron metabolism, lipid metabolism, and signaling pathways that regulate ferroptosis, has been implicated in the development and progression of OA, OP and OS (*Stockwell et al., 2017*; *Yang & Stockwell, 2016*). Furthermore, targeting specific ferroptosis-related factors has shown therapeutic benefits for these bone-related diseases (*Chen et al., 2023*). Hence, this article provides a comprehensive review of the role of ferroptosis in bone-related diseases, aiming to facilitate precise treatment through the targeted regulation of ferroptosis in the future.

### The intended audience

The primary intended audience for this review are researchers in the fields of orthopedics and/or cell biology. It provides a comprehensive explanation of the characteristics of ferroptosis and its connection between various bone-related diseases, particularly OP, OA, and OS, highlighting new perspectives on understanding the pathogenesis and potential therapeutic strategies for these diseases.

## SURVEY METHODOLOGY

We conducted a comprehensive literature search across PubMed, Google Scholar, and Web of Science using keywords 'ferroptosis,' 'osteoporosis,' 'osteoarthritis,' 'osteosarcoma,' and 'bone-related disease'. These terms were combined with Boolean operators to optimize the retrieval of relevant articles. Research, meta-analysis, editorials and review articles in the English language were included in this review. Letters and case report to the editor and case report were not included. A total of 129 articles published from November 1995 to April 2024 were selected for inclusion in this review.

### Features of ferroptosis

Ferroptotic cells display discernible alterations in their morphological, biochemical, and genetic features (*Dixon et al., 2012*). Morphologically, ferroptotic cells are characterized by increased cell volume, increased membrane density and rupture of the plasma membran, which are not observed in apoptotic cells (*Dixon et al., 2012*). In addition, mitochondrial ultrastructural abnormalities are also morphological features of ferroptosis, including reduced mitochondrial volume, increased membrane density, and absence of mitochondrial cristae (*Dixon et al., 2012*; *Yagoda et al., 2007*).

Ferroptotic cells exhibit detectable biochemical changes, with a dysregulated antioxidant system serving as a primary trigger for ferroptosis (*Dixon & Stockwell, 2014*). Lipid peroxidation occurs when oxidants, such as free radicals or ROS, attack the carbon-carbon double bonds of lipids (*Ayala, Muñoz & Argüelles, 2014*). Polyunsaturated fatty acids (PUFAs), which contain multiple carbon-carbon double bonds, are particularly susceptible to lipid peroxidation, making them key substrates in this process (*Ayala, Muñoz & Argüelles, 2014*). There are three phases to the lipid peroxidation process:
commencement, proliferation, and termination. In the commencement phase, carbon-centered lipid radicals are formed through the rearrangement of carbon radical molecules. During the proliferation phase, carbon-centered lipid radicals further transform into lipid hydroperoxides, the primary byproduct of lipid peroxidation. In the termination phase, non-radical products are formed (*Ayala, Muñoz & Argüelles, 2014*). In addition to lipid hydroperoxides, other byproducts of lipid peroxidation produced during ferroptosis include hexanal, propionaldehyde, malondialdehyde (MDA), and 4-hydroxynonenal (4-HNE) are produced during ferroptosis (*Ayala, Muñoz & Argüelles, 2014*). MDA and 4-HNE serve as dependable indicators of lipid peroxidation induced by oxidative stress in many diseases including cancer, Alzheimer's disease, and acute lung injury (*Park et al., 2021*; *Yang et al., 2022*).

The generation of lipid peroxides can compromise the structural integrity of the mitochondrial membrane, leading to its swelling. This swelling may trigger the irreversible opening of the mitochondrial permeability transition pore, causing a collapse in mitochondrial membrane permeability. As the outer mitochondrial membrane becomes more permeable, cytochrome c is released into the cytoplasm, leading to cell membrane contraction and rupture, ultimately resulting in cell death (*Bock & Tait, 2020*; *Bou-Teen et al., 2021*).

Iron metabolism dysregulation is another biochemical feature of ferroptosis. Accumulation of iron ions was observed during ferroptosis. The accumulation of iron ions leads to the excessive generation of ROS, which in turn promotes the formation of lipid hydroperoxides (*Yin, Xu & Porter, 2011*). The process of lipid peroxidation during ferroptosis is dependent on the accumulation of iron (*Liang, Minikes & Jiang, 2022*).

## Major pathways regulating ferroptosis

Ferroptosis is regulated by multiple pathways, and current research primarily focuses on GPX4 (glutathione peroxidase 4)-mediated pathway, FSP1-related pathways, tetrahydrobiopterin (BH4)-related pathways. Additionally, iron metabolism and PUFA metabolism are also involved in the regulation of ferroptosis (*Ayala, Muñoz & Argüelles, 2014*).

### Research on GPX4 in ferroptosis

GPX4, a member of the GPX family, is under intense scrutiny for its distinctive antioxidant function against membrane lipid peroxidation (*Liu et al., 2023*). GPX4 is the first discovered regulatory factor of ferroptosis (*Yang et al., 2014*). *Friedmann Angeli et al. (2014)* investigated the mechanisms of cell death in GPX4 knock-out mice and observed that lipid peroxidation occurred after GPX4 depletion. Additionally, it has been found that GPX4 can negatively regulate the expression of ROS, thereby inhibiting cell death (*Alim et al., 2019*; *Sengupta et al., 2013*). *Yang et al. (2014)* discovered that the manipulation of GPX4 can regulate cell death caused by 12 distinct ferroptosis inducers, but it does not affect inducers of other RCD forms. These observation demonstrates that GPX4 can inhibit lipid peroxidation and acts as an inhibitor of ferroptosis.

The GPX4 pathway is primarily composed of GPX4 and system Xc$^-$. GPX4 is the antioxidant enzyme that reduces lipid hydroperoxides. However, GPX4 alone only exhibits low enzymatic activity, its full functionality depends on the availability of cofactors like glutathione (GSH) (*Dixon et al., 2014*; *Stockwell & Jiang, 2020*). The system Xc$^-$ transporter plays a crucial role in maintaining cellular GSH levels by functioning as a cystine/glutamate antiporter, facilitating the uptake of cystine into cells in exchange for glutamate (*Dixon et al., 2014*; *Stockwell & Jiang, 2020*). SLC7A11 is a key component of the system Xc$^-$ transporter, responsible for mediating the import of cystine into the cell. Once inside the cell, cystine is converted to cysteine, which is then further synthesized into GSH. GPX4 facilitates the conversion of GSH to GSSH by scavenging lipid peroxides, thereby inhibiting ferroptosis (Fig. 1; *Liu et al., 2022*). Inhibiting GPX4 directly, as well as depleting GSH through small molecules like erastin or by deleting SLC7A11, can both induce ferroptosis (*Liu et al., 2022*).

### Research on FSP1-mediated pathway in ferroptosis

While GPX4 is a crucial factor in the inhibition of ferroptosis, it is noteworthy that in certain cell lines, inhibiting GPX4 does not necessarily lead to the induction of ferroptosis (*Doll et al., 2019*). This suggests that there are alternative pathways, independent of GPX4, that can also regulate the process of ferroptosis (*Bersuker et al., 2019*; *Doll et al., 2019*). Through studies on anti-ferroptosis cell lines, it was found that overexpression of FSP1 (also known as Apoptosis-Inducing Factor Mitochondrial-related 2, AIFM2) can alleviate cell death caused by GPX4 deletion or inactivation (*Doll et al., 2019*; *Ohiro et al., 2002*), suggesting that FSP1 is a ferroptosis regulatory factor that is independent of GPX4.

The mechanism by which FSP1 inhibits ferroptosis involves catalyzing the conversion of CoQ10 to its hydroquinone form, CoQ10H2 (*Zeng, Chen & Deng, 2022*). As a lipophilic antioxidant, CoQ10H2 effectively scavenges free radicals, thereby inhibiting lipid peroxidation (*Zeng, Chen & Deng, 2022*). In addition, CoQ10H2 can also indirectly promote the regeneration of another antioxidant, α-tocopherol, which inhibits the occurrence of ferroptosis by capturing free radicals (Fig. 1; *Chen et al., 2022*; *Doll et al., 2019*). The myristoylation (a type of post-translational modification) of FSP1 is necessary for its role in regulating ferroptosis, as it facilitates the localization of FSP1 to cellular membranes where it can interact with CoQ10 and other lipid components (*Bersuker et al., 2019*). This indicates a direct association between the FSP1-CoQ10 pathway and the process of ferroptosis.

### Research on GCH1-BH4 pathway in ferroptosis

BH4, a redox-active factor, possesses antioxidant properties and plays a crucial role in maintaining redox balance (*Soula et al., 2020*). The *de novo* synthesis of BH4 commences with guanosine triphosphate (GTP), with GTP cyclohydrolase-1 (GCH1) serving as the key enzyme in this process (*Kojima et al., 1995*; *Soula et al., 2020*). Two independent research groups demonstrated that GCH1-BH4 is a new pathway for regulating ferroptosis (*Kraft et al., 2020*; *Soula et al., 2020*; Fig. 1). GCH1 overexpression not only abolished lipid peroxidation but also protected against RSL3 (a GPX4 inhibitor)-induced ferroptosis or

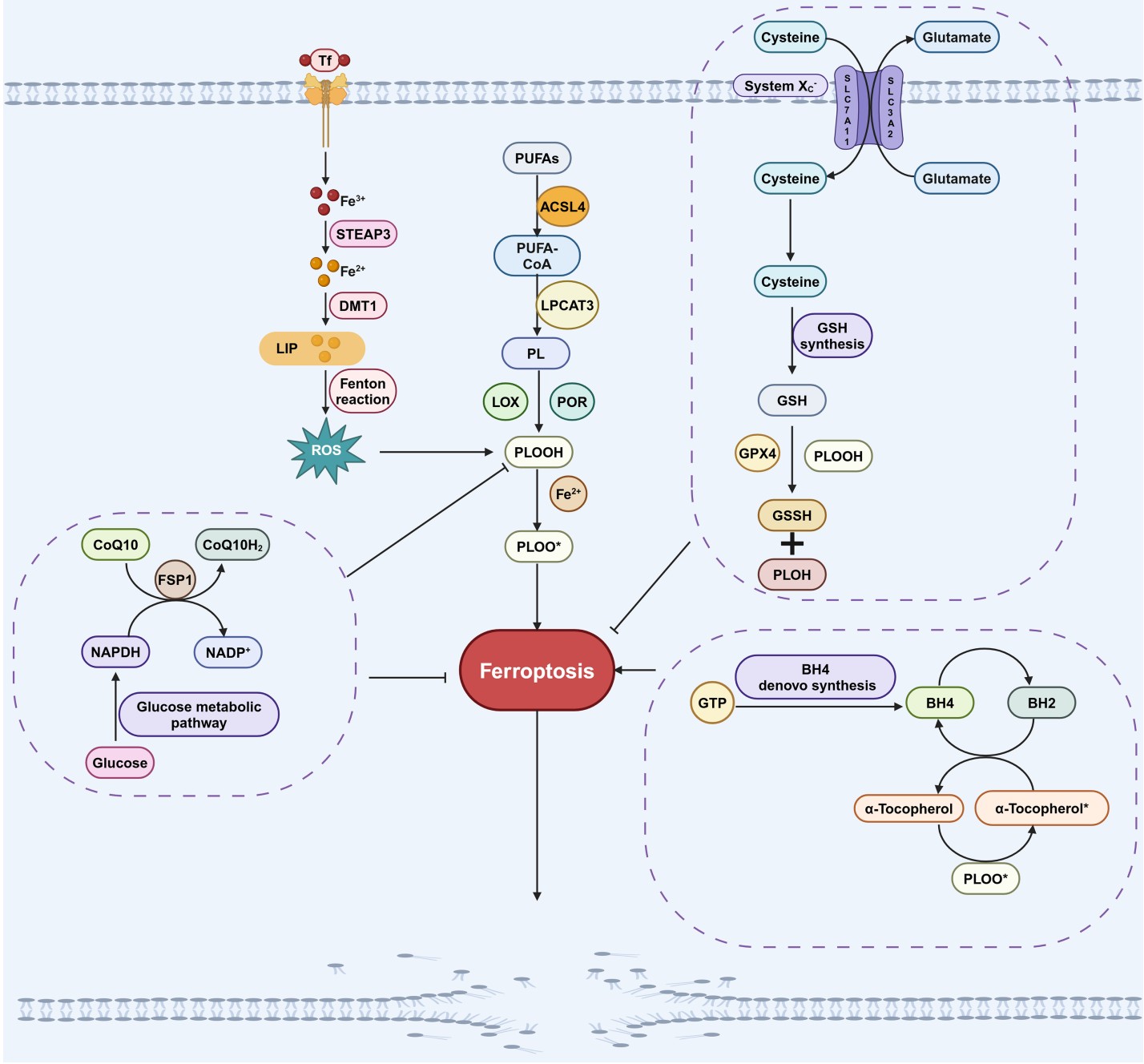

**Figure 1 Pathways involved in regulating ferroptosis.** The GPX4 pathway, NADPH-FSP1-CoQ10 pathway, GCH1-BH4 pathway, iron metabolic pathways and lipid peroxidation pathway are the main pathways regulating ferroptosis. The GPX4 pathway inhibits ferroptosis by consuming PLOOH and suppressing the generation of lipid peroxides. Iron can promote the production of ROS, which in turn promotes the production of PLOOH, thereby promoting ferroptosis. CoQ10 is converted into its reduced form CoQ10H2 under the action of FSP1, which then inhibits the formation of PLOOH and thus suppresses ferroptosis. PUFAs can generate PLOOH through a series of biological processes, promoting the occurrence of ferroptosis. During the conversion of BH4 to BH2, PLOOH can be produced, which promotes ferroptosis. These pathways will eventually affect phospholipid hydroperoxides (PLOOH), thereby affecting the occurrence of ferroptosis. Created in BioRender.

GPX4 deletion-induced ferroptosis. However, GCH1 overexpression cannot reverse other forms of RCD including apoptosis and necrosis (*Kraft et al., 2020*), indicates that GCH1 can selectively antagonize ferroptosis. Additionally, the administration of BH4 to cells treated with ferroptosis inducers effectively rescued cells from undergoing ferroptosis. These findings underscore the GCH1-BH4 pathway's pivotal role in inhibiting ferroptosis.

### Research on iron metabolism in ferroptosis

Iron must undergo a series of absorption and metabolic processes before it can participate in the regulation of ferroptosis. Iron ($Fe^{3+}$) absorbed by the small intestine is transported to the cell membrane *via* transferrin (Tf) and then enters the cell with the assistance of transferrin receptor 1 (TfR1). Subsequently, $Fe^{3+}$ is converted to $Fe^{2+}$ by STEAP3 (six-transmembrane epithelial antigen of prostate 3). Then, $Fe^{2+}$ is conveyed by the divalent metal transporter 1 (DMT1) to produce ferritin (FN) and the labile iron pool (LIP) (*Miao et al., 2023*; *Torii et al., 2016*). LIP produces a large number of hydroxyl radicals through Fenton reaction under pathological conditions. This in turn results in an increased production of ROS and ultimately contributes to the occurrence of ferroptosis (*Kajarabille & Latunde-Dada, 2019*; Fig. 1).

### Research on PUFA metabolism in ferroptosis

As mentioned above, PUFAs are the primary targets of ROS and the main substrates for the formation of lipid peroxides, making the regulation of PUFAs crucial for the process of ferroptosis (*Gao et al., 2023*; *Zheng, Liang & Zhang, 2023*). In the catalytic process, PUFAs initially form a complex with coenzyme A (CoA) under the catalyzation of acetyl-CoA synthetase long-chain family member 4 (ACSL4). Subsequently, this complex further catalyzed by lysophosphatidylcholine acyltransferase 3 (LPCAT3), leading to the formation of membrane phospholipids, which are the key signals for the stimulation of ferroptosis (*Qi & Peng, 2023*; Fig. 1). Inhibition of ACSL4 and LPCAT3 inhibits ferroptosis through multiple pathways (*Kagan et al., 2017*). Conversely, *Lei et al. (2020)* activated ACSL4 through ionizing radiation, leading to the occurrence of ferroptosis. Hence, the lipid peroxidation process facilitated by ACSL4 and LPCAT3 represents a significant pathway eliciting ferroptosis.

## The relationship between ferroptosis and bone-related diseases
### The link between ferroptosis and OP

OP, a metabolic bone disease, is characterized by an imbalance between the functions of osteoblasts and osteoclasts in bone formation and resorption (*Hernlund et al., 2013*). This imbalance results in bone mass loss and reduced bone strength, increasing the risk of brittle fractures and hindering fracture healing (*Yamamoto et al., 2021*). About 8.9 million people in the world suffer from osteoporotic fractures, among which hip and spine fractures caused by osteoporosis have become a major global public health problem (*Hernlund et al., 2013*; *Sözen, Özışık & Başaran, 2017*). It is estimated that the incidence of hip fractures will increase by 310% in older men and 240% in older women in 2050 (*Bass et al., 2019*). Therefore, preventing the occurrence and progression of OP has emerged as a matter of paramount global importance. Studies indicate that ferroptosis may regulate the

occurrence and development of OP by affecting osteoblast and osteoclast functions through influencing bone metabolism-related signaling pathways (*Yan et al., 2022a*).

*Ferroptosis and diabetic osteoporosis (DOP)*

About 9% of adults worldwide are affected by diabetes mellitus, and this number has undergone a four-fold increase over the past three decades (*Zheng, Ley & Hu, 2018*). There is a close relationship between diabetes and OP, two metabolic diseases (*Anagnostis et al., 2018*). Oxidative stress and the accumulation of advanced glycation end products (AGEs) in collagen, triggered by hyperglycemia, are two significant factors contributing to reduced bone formation (*Anagnostis et al., 2018*; *Farr et al., 2014*). Other studies have shown that low levels of insulin and insulin-like growth factor-1(IGF-1) may affect the osteogenesis of osteoblasts, which in turn leads to the decrease of bone strength (*Epstein & LeRoith, 2008*). Two independent experimental groups showed that some diabetes drugs, such as thiazolidinediones, would increase the risk of fracture in diabetic patients (*Eller-Vainicher et al., 2020*; *Napoli et al., 2017*). The complex pathogenic mechanisms underlying diabetic osteoporosis are not yet fully understood; however, research has revealed that the dysregulation of ferroptosis contributes to its development.

Iron metabolism is often disturbed in diabetes (*Liu et al., 2020*). The dysregulation of factors involved in iron metabolism, such as transferrin, ferritin, hepcidin, and the transferrin receptor were observed on the onset and progression of type 2 diabetes (*Liu et al., 2020*). *Wang et al. (2022b)* identified several ferroptosis-related phenomenon in a type 2 diabetes mellitus (T2DOP) rat model, including increased in ROS level, accumulation of lipid peroxides, and decreased expression of ferroptosis regulator including GPX4. Further investigation showed that the expression of mitochondrial ferritin (FtMt), a protein involved in storing iron ions, was elevated in the bone tissue of T2DOP rats. Overexpression of FtMt mitigated oxidative stress, suppressed the occurrence of ferroptosis and improved the functionality of osteoblasts high glucose condition. Conversely, knock-down of FtMt hampered the osteogenic function in a ferroptosis-dependent manner, highlighting the significant role of FtMt-mediated ferroptosis in the development T2DOP. Under high-glucose conditions, MC3T3-E1 osteoblasts exhibit activation of ferroptosis characteristics, including an increase in lipid peroxidation products, abnormal changes in mitochondrial structure, and a decrease in GPX4 expression levels. These changes ultimately lead to impaired cell viability and inhibit their osteogenic differentiation capacity (*Ma et al., 2020*). Melatonin is a potent endogenous antioxidant. It can reduce ferroptosis induced by high glucose by activating the Nrf2/HO-1 signaling pathway, thereby promoting the osteogenic ability of MC3T3-E1 cells (*Ma et al., 2020*).

In conclusion, the disruption of osteoblast iron homeostasis and the dysregulation of ferroptosis are closely linked to the progression of T2DOP, and ferroptosis-related pathways hold promise as potential therapeutic targets for the treatment of T2DOP (Fig. 2).

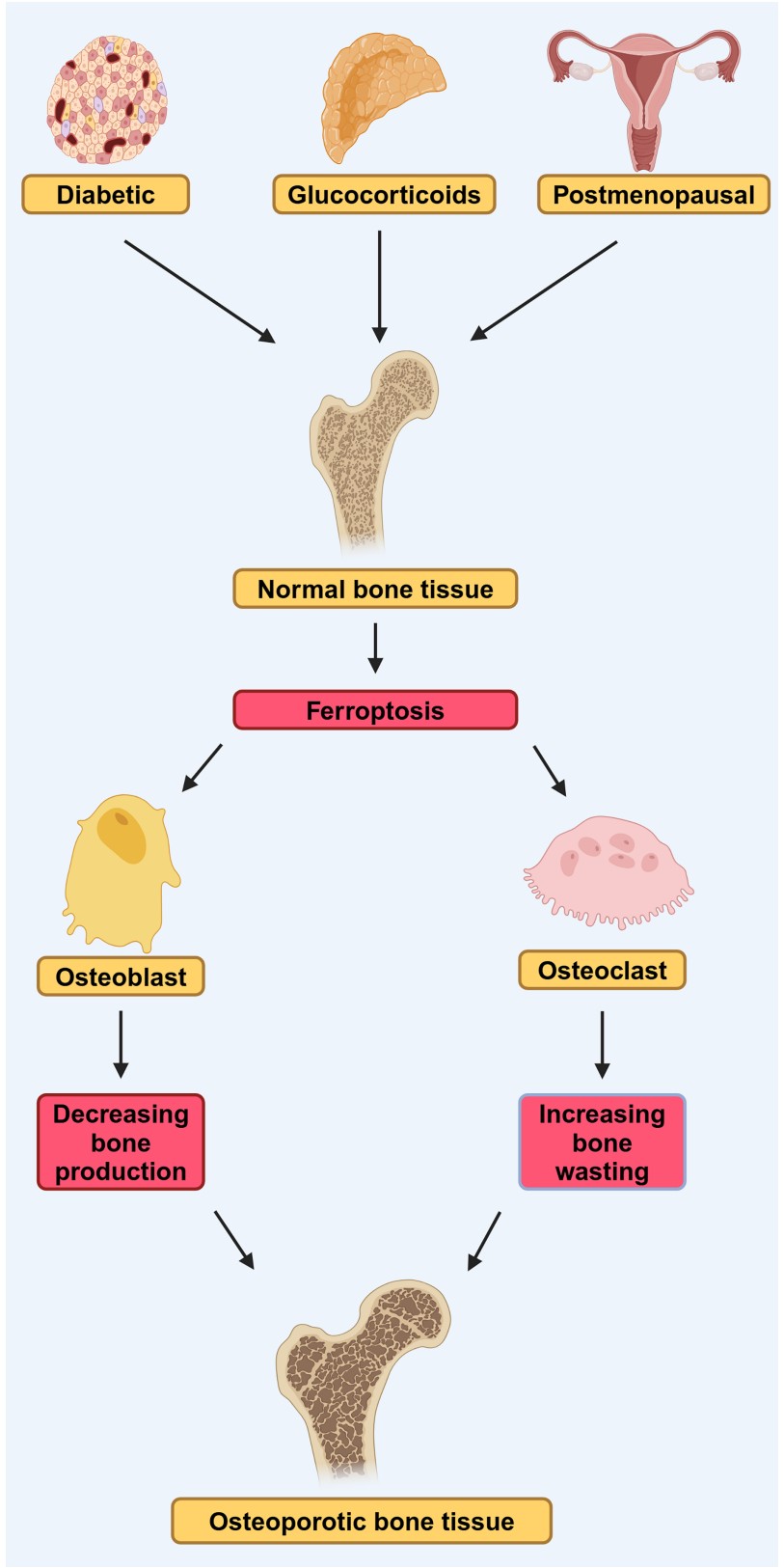

**Figure 2 DOP, GIOP, and postmenopausal osteoporosis (PMOP) in osteoporosis induced by ferroptosis.** Diabetes, estrogen, and glucocorticoids can affect osteoblasts and osteoclasts through
**Figure 2** (continued)
ferroptosis, leading to osteoporosis. They lead to reduced bone-forming capacity of osteoblasts and reduced bone-clearing capacity of osteoclasts, by affecting ferroptosis (modified from *Liu et al., 2022*). Created in BioRender.                         

*Ferroptosis and glucocorticoid-induced osteoporosis*

Ferroptosis has recently been associated with the pathogenesis of glucocorticoid-induced osteoporosis (GIOP) (Fig. 2). Prolonged administration of high-dose steroid hormones, such as glucocorticoid, can impair the body's antioxidant capacity, subsequently diminishing the activity of osteoblasts and contributing to the onset of OP (*Yang et al., 2021*). *Lu et al. (2019)* found that high-dose dexamethasone (10 µM) down-regulated GPX4 and system $Xc^-$, which led to ferroptosis. Further bioinformatics analysis confirmed that ferroptosis pathway was activated after high-dose dexamethasone was administered, indicating that ferroptosis play an important role in the development the GIOP.

Furthermore, evidence suggests that targeting ferroptosis is an effective strategy for alleviating GIOP. *Yang et al. (2021)* found that endothelial cell-secreted exosomes (EC-Exos), which serve as a crucial mediator for intercellular communication, can reverse the cortical hormone-induced osteogenic inhibition by inhibiting ferroptosis *in vitro* and *in vivo*. Treatment of mice with extracellular vesicles derived from bone marrow endothelial progenitors (EPC-EVs) inhibits the activation of ferroptosis signaling and augments skeletal parameters impacted by high-dose dexamethasone, indicating that EPC-EVs hold promise for future clinical treatment of GIOP.

*Ferroptosis and postmenopausal osteoporosis (PMOP)*

Estrogen deficiency in postmenopausal women impairs osteoblast differentiation and increases osteoclast activity, resulting in osteopenia and greater bone brittleness, which are the primary contributors to PMOP (*Eastell, 1998*). *Okyay et al. (2013)* found that there was also a significant correlation between low serum iron level and PMOP. Iron can affect the functions of osteoblasts and osteoclasts through various pathways, thereby influencing OP. Iron can induce autophagy, apoptosis, and ferroptosis in osteoblasts, thereby promoting the occurrence of OP. It can also promote the production of osteoprotegerin (OPG) to enhance the function of osteoclasts, which also contributes to the progression of OP (Fig. 3; *Cai et al., 2023*). Research has found that reduction of ferritin promoted RANKL-induced ferroptosis of osteoclasts (*Ni et al., 2021*). The hypoxia-inducible-factor-1-alpha (HIF-1α) helps to reduce ferritin autophagy under hypoxia, thereby promoting ferroptosis in osteoclasts and advancing the progression of OP. This process can be reversed by the HIF-1α inhibitor 2-methoxyestradiol (2ME2) (*Ni et al., 2021*). These findings suggest that an alternative therapeutic approach for OP could be attained by targeting HIF-1α and ferritin to modulate ferroptosis in osteoclasts.

The current research results strongly indicate the widespread presence of disrupted iron metabolism, oxidative stress, and lipid peroxidation pathways that ultimately lead to ferroptosis in individuals with OP (Fig. 2). Studies have shown that some drugs can target ferroptosis and exhibit positive effects in the treatment of. 1,25-dihydroxyvitamin D3, as

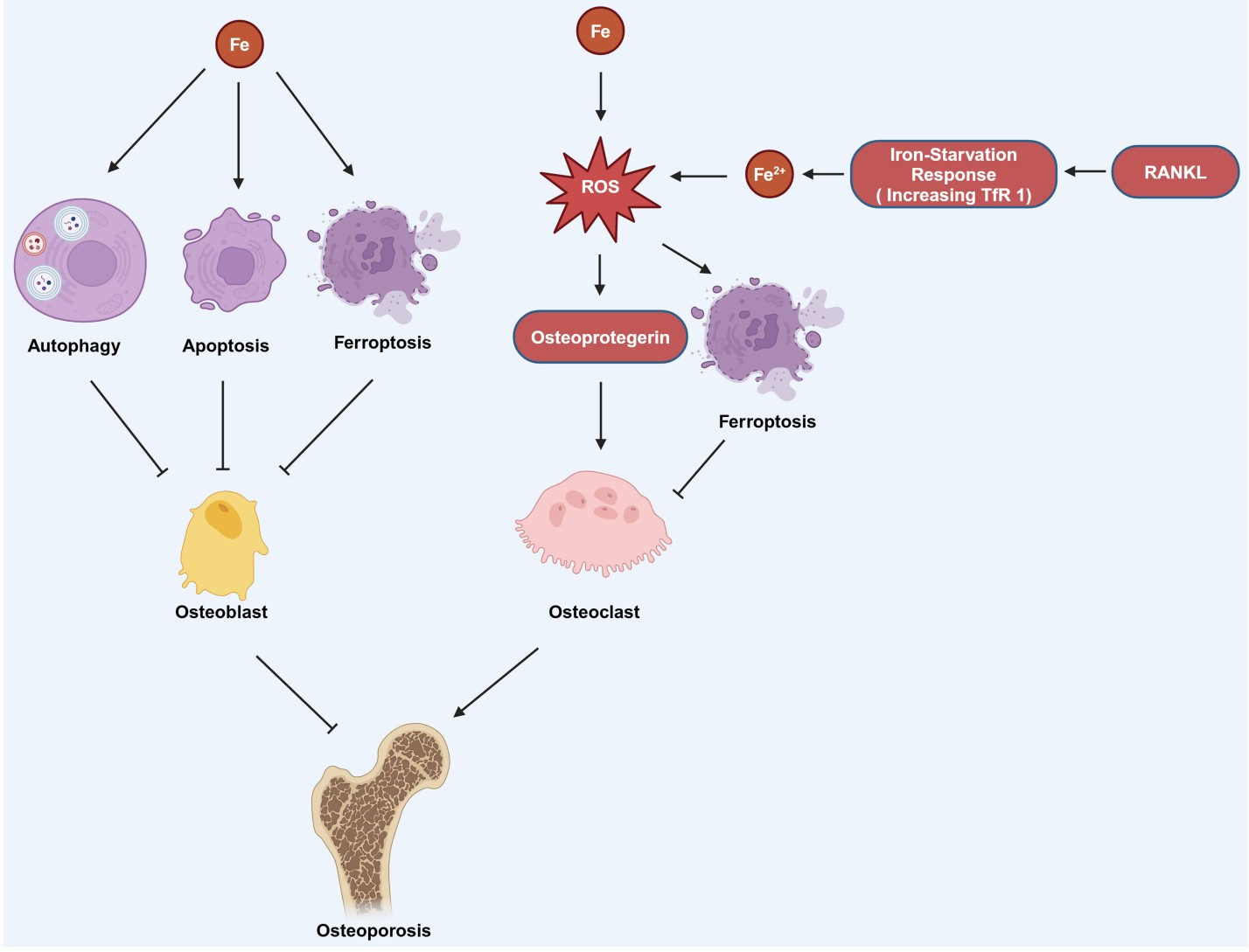

**Figure 3** **The mechanism by which iron affects OP involves several pathways.** Iron can induce autophagy, apoptosis, and ferroptosis in osteoblasts, which contributes to the development of OP. Additionally, iron can stimulate the production of ROS, which in turn can increase the production of osteoprotegerin. Elevated osteoprotegerin levels enhance the activity of osteoclasts, the cells responsible for bone resorption, thereby promoting the occurrence and progression of OP. Moreover, the generation of ROS can also lead to ferroptosis in osteoclasts. RANKL can induce iron starvation response (increased TfR1) leading to increased intracellular $Fe^{2+}$, promoting ROS production, and facilitating ferroptosis. Created in BioRender.

an agonist of the vitamin D receptor, can promote the expression of Nrf2 and inhibit ferroptosis in osteoblasts after activating vitamin D receptor, thereby delaying OP (*Xu et al., 2022b*). Similarly, the Qing' pill inhibited ferroptosis in osteoblasts and promotes osteoblast osteogenesis by promoting the AKT/PI3K pathway, and also achieved the effect of delaying the occurrence of OP (*Hao et al., 2022*). In addition, melatonin can effectively reduce the expression of ferroptosis-related proteins in bone marrow mesenchymal stem cells, thereby inhibiting their ferroptosis and also delaying the occurrence of OP (*Li et al., 2022*). Thus, the modulation of ferroptosis in osteoclasts or osteoblasts may represent an innovative therapeutic approach for the management of OP (Table 1).

**Table 1  Effects of different compounds or biological factors on ferroptosis regulation in OP.**

| Factor | Mechanism | Effect | Reference |
|---|---|---|---|
| Carbonyl cyanide m-chlorophenyl-hydrazine (CCCP) | Promoting osteoblast ferroptosis by activating mitochondria | Accelerating OP | Wang et al. (2022b) |
| Melatonin | Inhibiting ferroptosis in MC3T3-E1 by activation of Nrf2/HO-1 signaling | Preventing OP | Ma et al. (2020) |
| Endothelial cell-secreted exosomes (EC-Exos) | Reversal of the inhibitory effect of glucocorticoids on osteoblasts by inhibiting ferritin phagocytosis-dependent ferroptosis | Preventing OP | Yang et al. (2021) |
| Extracellular vesicles extracted from bone marrow-derived endothelial progenitor cells (EPC-EVs) | Inhibiting of ferroptosis pathway in osteoblasts by restoring Gpx4 and system Xc- level | Preventing OP | Lu et al. (2019) |
| 2-methoxyestradiol (2ME2) | Inducting of osteoclast ferroptosis by targeting hypoxia-inducible factor 1α and ferritin | Preventing OP | Ni et al. (2021) |
| 1,25-dihydroxyvitamin D3 (1,25-(OH)2D3) | Inhibiting ferroptosis of osteoblasts by promoting Nrf2 expression | Preventing OP | Xu et al. (2022b) |
| Qing' e Pill | Inhibiting ferroptosis in osteoblasts by promoting the AKT/PI3K pathway | Preventing OP | Hao et al. (2022) |
| Gastrodin | Protection of osteoblast activity by inhibiting osteoblast ferroptosis through anti-oxidation | Preventing OP | Li & Li (2022) |
| Zoledronic | Promoting osteoclast ferroptosis by increasing the stability of p53 protein | Accelerating OP | Qu et al. (2021) |

### The link between ferroptosis and OA

In 2006, Dombrecht et al. (2006) initially observed that the addition of $Fe^{2+}$ enhanced lipid peroxidation in chondrocytes treated with tert-butyl hydroperoxide (t-BHP) or hydrogen peroxide. In recent years, the link between OA, dysregulated iron metabolism, and lipid peroxidation has attracted increasing attention. Jing et al. (2021a) found that up-regulation of TfR1 and down-regulation of FPN by Interleukin-1 beta (IL-1β) and Tumor necrosis factor-α (TNF-α) can disrupt iron homeostasis in chondrocytes (Fig. 4), revealing that oxidative stress and disrupted iron homeostasis in chondrocytes contribute to the development of osteoarthritis.

An increasing amount of data suggests that the GPX4-mediated ferroptosis pathway is involved in the development of OA. Mo, Xu & Li (2021) found that IL-1β-treated mouse chondrocytes (ATDC5) had increased intracellular $Fe^{2+}$ and MDA levels, alongside elevated expression of TfR1, and decreased expression of GPX4 and SLC7A11. Similarly, Yao et al. (2021) discovered that chondrocytes treated with IL-1β to induce inflammation exhibit an accumulation of lipid-derived reactive oxygen species (lipid-ROS), decreased GPX4 expression, and subsequently impaired the functionality of chondrocytes. These findings revealed that the GPX4-mediated pathway was down-regulated, leading to the activation of ferroptosis in OA. Further evidence showed that inhibition of ferroptosis using chemical compound Ferrostatin-1 attenuated cartilage degradation in OA mouse model. Similarly, Guo et al. (2022) discovered administration of deferoxamine (DFO), an effective iron chelator, inhibited ferroptosis activation and showed protective effects on

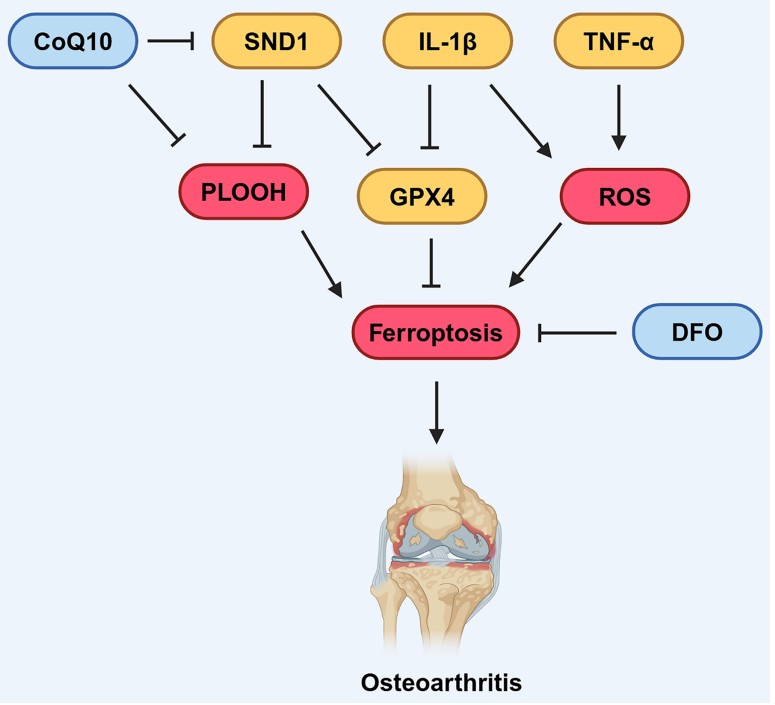

**Figure 4 Various factors influence ferroptosis in OA.** CoQ10 can inhibit the formation of lipid peroxides, thereby suppressing ferroptosis. DFO can also suppress ferroptosis. SND1 and IL-1β can inhibit the expression of GPX4, thus promoting ferroptosis. IL-1β and TNF-α can promote the production of ROS, which in turn promotes ferroptosis. Created in BioRender.

both osteoarthritic chondrocyte model and OA mouse model, demonstrating that inhibiting ferroptosis has beneficial effects on OA (Fig. 4).

Several pathways have been identified that involved in GPX4-mediated ferroptosis in OA models. *Lv et al. (2022)* also reported that the intracellular $Fe^{2+}$ and MDA levels increased, while the expression of GPX4 decreased after inflammation activation. Mechanistically, they also found that the RNA-binding protein SND1 inhibited the expression of heat shock protein family A family member 5 (HSPA5), a factor that promotes the degradation of GPX4 (*Lv et al., 2022*). This indicates that the SND1-HSPA5 pathway acts as an up-stream inhibitory regulator of GPX4 expression, thereby contributing to the promotion of ferroptosis in osteoarthritic chondrocytes (Fig. 4; *Lv et al., 2022*). Additionally, D-mannose has been reported to protect osteoarthritic chondrocytes by reducing their susceptibility to ferroptosis. *Zhou et al. (2021)* found that D-mannose administration significantly suppressed the expression of HIF-2α. Further, they demonstrated that silencing HIF-2α promoted the expression of GPX4, while overexpression of HIF-2α diminished the protective effect of D-mannose against ferroptosis. This evidence suggests that D-mannose-mediated suppression of HIF-2α acts as an upstream negative regulator of GPX4-mediated ferroptosis in osteoarthritic chondrocytes.

It has also been demonstrated that other ferroptosis pathways, such as the FSP1/CoQ10 axis, have significant regulatory roles in OA. *Chang et al. (2020)* conducted a case-control study comparing plasma CoQ10 level in patients with osteoarthritis and non-osteoarthritis patients and found that the CoQ10 level are slightly lower in OA patients, suggesting a negative correlation between OA and plasma CoQ10. In another study using an OA rat model, *Lee et al. (2013)* found that CoQ10 exhibited therapeutic benefits for OA, including pain alleviation and the prevention of cartilage degeneration. These results suggest a potential role of CoQ10 in preventing degradation of cartilage in OA patients.

As research into the role of ferroptosis in OA advances, the management of lipid peroxidation and iron homeostasis has emerged as two promising new approaches for OA treatment (*Zhang et al., 2022*). For instance, DFO (*Burton et al., 2022*; *Miao et al., 2022*), antioxidants such as Fer-1 (*Miao et al., 2022*; *Zhou et al., 2021*) and CoQ10 (*Lee et al., 2013*; *Li et al., 2017*) have demonstrated significant anti-OA effects in both *in vivo* and *in vitro* studies. In addition, the calcium chelator BAPTA acetoxymethyl ester (BAPTA-AM) (*Jing et al., 2021b*), the lactoferrin (*Rasheed, Alghasham & Rasheed, 2016*), and the stigmasterol (*Mo, Xu & Li, 2021*) have also shown anti-OA effects (Table 2). At the same time, *Wang et al. (2022a)* and *Gong et al. (2023)* found that astaxanthin that is a type of xanthophyll carotenoid known for its anti-inflammatory and antioxidant characteristics and cardamonin that is a compound found in the ginger family, known for its antioxidant and anti-inflammatory properties inhibited ferroptosis of chondrocytes induced by IL-1β, thereby delaying the progression of the disease. *Xu et al. (2022a)* and *Yan et al. (2022b)* respectively found that metformin and theaflavin-3,3-digallate can also inhibit ferroptosis of chondrocytes, contributing to the alleviation of OA progression. These ferroptosis-targeting therapeutic approaches offer a broader range of options for the future OA treatment; however, further clinical research is essential to evaluate the effectiveness and safety of these agents.

### The link between ferroptosis and OS

RCD such as apoptosis and necrosis exert substantial influence on the progression of OS and can significantly modulate its clinical prognostic outcome (*Li et al., 2016*). As a recently discovered form RCD, ferroptosis has been gradually elucidated in its connection to OS. Evidence has shown that the expression of genes involved in ferroptosis regulation were dysregulated in OS, and inducing ferroptosis in OS cells can result in marked cytotoxicity, including reduced cell viability and increased cell death (*Lin et al., 2021*), indicating ferroptosis is a potential therapeutic target for OS.

Several signaling pathways have been shown to regulate the progression of OS by targeting ferroptosis. The signal transducer and activator of transcription 3 (STAT3) signaling pathway is associated with the development of OS and its activation promotes immune evasion, drug resistance, and distant metastasis of OS cells (*Guanizo et al., 2018*; *Liu et al., 2021*). As a down-stream effector of the STAT3 pathway, Nuclear factor erythroid 2-related factor 2 (Nrf2) has been established to promote the expression of GPX4 in the context of cancer (*Gao et al., 2017*; *Shin et al., 2018*). The expression of STAT3, Nrf2 and GPX4 in drug-resistant OS cell lines increased compared with non-drug-resistant lines

**Table 2 Effects of different compounds on the regulation of ferroptosis in OA.**

| Compounds | Mechanism | Effect | Reference |
|---|---|---|---|
| Calcium chelator BAPTA acetoxymethyl ester (BAPTA-AM) | Reducing iron levels and ROS production in chondrocytes | Delaying the occurrence of OA | *Jing et al. (2021b)* |
| Nifedipine | Reducing the production of ROS through the NrF2 pathway | Delaying the occurrence of OA | *Yao et al. (2020)* |
| Icariin | Reversing the matrix degradation effects of IL-1β and reducing ROS production | Delaying the occurrence of OA | *Zuo et al. (2019)* |

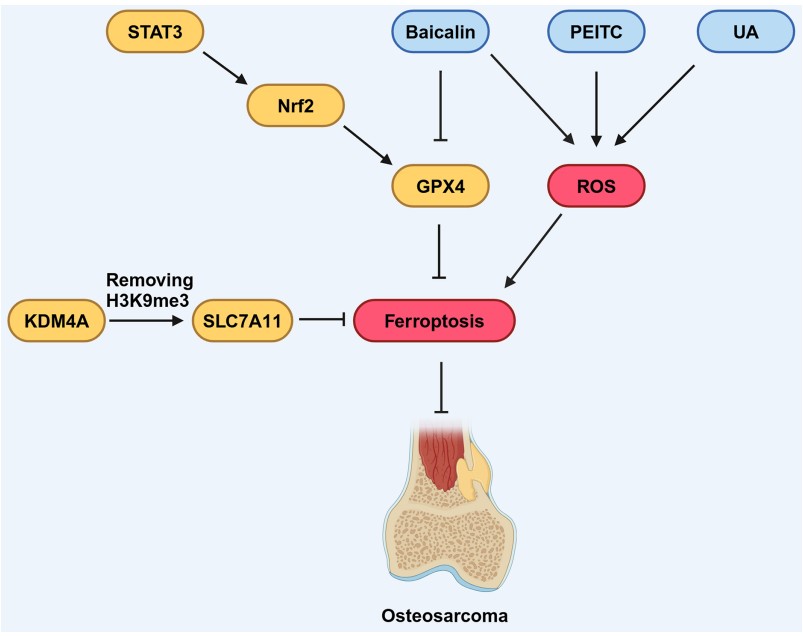

**Figure 5 Various factors influence ferroptosis in OS.** STAT3 can promote the expression of downstream molecule Nrf2, which in turn promotes the expression of GPX4, thereby inhibiting ferroptosis. Baicalin, PEITC, and UA can promote the production of ROS, thus promoting ferroptosis. Baicalin can also inhibit the expression of GPX4, contributing to the promotion of ferroptosis. KDM4A can remove the H3K9me3 modification, thereby promoting the expression of SLC7A11, which inhibits ferroptosis. Created in BioRender.

(*Liu & Wang, 2019*). STAT3 inhibition increased ROS generation, reactivate ferroptosis of OS cells and increased its sensitivity to chemotherapy drug treatment, such as 5-fluorouracil. These findings underscore the potential of the STAT3 signaling pathway as a promising regulator of ferroptosis in the context of OS therapy (*Cai et al., 2017*; *Zuo et al., 2017*; Fig. 5).

In addition, other signaling pathways, including the MAPK signaling pathway and the HIF signaling pathway, which can regulate the production of ROS, have been implicated in the regulation of OS development related to ferroptosis (*Fu et al., 2021*; *Lv et al., 2020b*). However, the intricate molecular mechanisms underlying this connection remain elusive.

Ferroptosis in OS is modulated by components that have been identified as participants in epigenetic regulation. *Chen, Jiang & Sun (2021)* noted that KDM4A, a histone
demethylase, exhibited increased expression levels in OS samples. Furthermore, they found high KDM4A expression was associated with a poor prognosis for OS patients, implicating KDM4A as an oncogene in the OS. Notably, the knock-down of KDM4A was found to enhance ferroptosis and inhibit the progression and metastasis of OS cells. Mechanistically, KDM4A promotes the transcription of SLC7A11 by removing H3K9me3, a histone modification that suppresses gene expression, from its promoter region. Knockdown of KDM4A reduces SLC7A11 expression, leading to inhibition of the GPX4 pathway and activation of ferroptosis (Fig. 5).

Ferroptosis not only functions independently but has also been found to have substantial correlations with other types of RCD in a variety of malignancies, such as OS (*Koren & Fuchs, 2021*; *Woo et al., 2020*). An ultrasound-activatable nanomedicine for treating OS developed by *Fu et al. (2021)* inhibits tumor growth by inducing and synergizing ferroptosis and apoptosis, suggesting that ferroptosis can coexist and synergize with another mode of cell death in OS cells. In a related study by *Lv et al. (2020b)* OS cells from K7M2 mice treated with phenethyl isothiocyanate (PEITC) elicited various RCD, including ferroptosis, apoptosis, and autophagy (Fig. 5). When these RCD processes were reversed using specific inhibitors, it was observed that inhibitors targeting other RCD pathways only partially rescued cell survival, whereas the ferroptosis inhibitor almost entirely prevented cell death. This suggests that ferroptosis may have a dominant role when collaborating with various forms of RCD in OS.

Numerous studies have highlighted the efficacy of combining traditional Chinese medicine with conventional anti-tumor treatments in suppressing tumor growth, improving survival rates, and reducing side effects caused by conventional therapies, thereby offering substantial benefits to patients (*Liu et al., 2019*; *Tang et al., 2019*). Certain compounds derived from traditional Chinese medicine herbs, or their derivatives have demonstrated inhibitory effects on the growth of OS cells (*Liu et al., 2013*; *Wang, Fu & Zhao, 2013*). Among these compounds, Baicalin, a flavonoid compound extracted and isolated from the dried roots of Huang Qin (*Scutellaria baicalensis*), has demonstrated the capacity to regulate ferroptosis in various pathological conditions, including cancers (*Wen et al., 2023*; Fig. 5). Notably, Baicalin has been found to inhibit the growth of bladder cancer cells and OS cells by inducing ferroptosis (*Kong et al., 2021*; *Wen et al., 2023*; Table 3), suggesting its potential to exert anticancer pharmacological effects through the activation of ferroptosis. Similarly, ursolic acid (UA), a natural pentacyclic triterpenoid carboxylic acid in kiwifruit, can induce lipid peroxidation and induce iron sag in OS cells (*Tang et al., 2021*; Table 3; Fig. 5). Further research results indicating that traditional Chinese medicine combined with conventional chemotherapy or radiotherapy can improve the survival rate of OS patients (*Tang et al., 2021*). These findings underscore the close relationship between traditional Chinese medicine and ferroptosis, providing innovative approaches for OS treatment.

It is worth mentioning that chemotherapy is still the mainstream research direction of ferroptosis in the treatment of OS. In contrast, other treatment methods such as radiotherapy and immunotherapy, which have demonstrated strong associations with ferroptosis in the context of other solid tumors, have not been extensively reported in OS

**Table 3  Effects of different compounds on ferroptosis regulation in OS.**

| Compounds | Mechanism | Effect | Reference |
|---|---|---|---|
| Sulfasalazine | Inhibiting system Xc- | Promoting ferroptosis | Dixon et al. (2012) |
| EF24 | Laying up ROS | Promoting ferroptosis | Lin et al. (2021) |
| Nanomedicine | Laying up $Fe^{2+}$ and taking GSH | Promoting ferroptosis | Fu et al. (2021) |
| Baicalin | Inhibiting GPX4 and promoting ROS accumulation | Promoting ferroptosis | Wen et al. (2023) |
| Ursolic acid (UA) | Causing lipid peroxidation | Promoting ferroptosis | Tang et al. (2021) |
| Bavachin | Inhibiting GPX4 and SLC7A11 | Promoting ferroptosis | Luo et al. (2021) |
| phenethyl isothiocyanate (PEITC) | Taking GSH | Promoting ferroptosis | Lv et al. (2020a, 2020b) |
| Tirapazamine | Inhibiting SLC7A11 | Promoting ferroptosis | Shi et al. (2021) |
| Artemisinin | Influence $Fe^{2+}$ levels | Promoting cytotoxicity | Isani et al. (2019) |
| Liproxstatin-1 | Inhibiting ROS | Inhibiting ferroptosis | Li et al. (2019b) |
| Esomeprazole | Inhibiting ROS | Inhibiting ferroptosis | Ebrahimpour et al. (2021), Kajarabille & Latunde-Dada (2019) |

(Zhao et al., 2021). It has been confirmed that radiation therapy can down-regulate the SLC7A11, up-regulate ACSL4, activate the cGAS pathway, and affect various other pathways to modulate ferroptosis in a variety of cancer cells (Choi, Kipps & Kurzrock, 2016; Lang et al., 2019; Li et al., 2021). The relationship between ferroptosis and immunotherapy has been demonstrated in specific tumor cells, and the immunogenicity of ferroptosis has also been investigated in fibrosarcoma, which belongs to the category of musculoskeletal cancers, similar to OS (Dai et al., 2020; Efimova et al., 2020). Due to the limited evidence, the potential effects of ferroptosis in non-pharmacological treatments for OS remains to be definitively established (Zhao et al., 2021).

## CONCLUSION AND OUTLOOK

Ferroptosis is a recently identified form of RCD that has attracted a lot of attention lately because of its possible role in the etiology of many diseases. Significant progress has been made in studies in the last few years, especially in determining the connections between ferroptosis and bone-related diseases. Exploring the underlying mechanisms of ferroptosis in the context of OP, OA, and OS is providing valuable insights into the pathogenesis of these conditions and new perspectives for their treatment.

However, it is important to note that the precise mechanisms governing ferroptosis are not yet fully elucidated, these knowledge gaps pose challenges to research progress and clinical applications. Numerous signaling pathways, such as the MAPK signaling pathway and the HIF signaling pathway, are associated with ferroptosis. The activation of the MAPK pathway causes the generation of ROS, leading to speculation that the MAPK signaling pathway plays a role in regulating ferroptosis. Meanwhile, evidence has shown that lactate induced by HIF-1α suppresses ferroptosis in a pH-dependent manner, indicating that HIF-1 is potentially involved in the regulation of ferroptosis. However, the molecular mechanisms linking these pathways to ferroptosis and its roles in bone-related
diseases, especially in OS, are not yet clear. Therefore, further investigation is needed to elucidate the molecular links between these signaling pathways and ferroptosis, which could pave the way for new therapeutic strategies targeting ferroptosis in disease treatment.

The development and progression of bone-related diseases involve the interaction of multiple cell types, including osteoblasts, osteoclasts, mesenchymal stem cells and chondrocytes. However, the precise regulation of ferroptosis mechanisms in these cell types remains unclear. Whether key ferroptosis regulators, such as GPX4, SLC711A and FSP1, show similar or distinct expression profiles in different cell types within bone tissue under disease conditions is not yet fully understood. Additionally, the functions and downstream mechanisms of these ferroptosis regulators in various cell types within bone tissue are not yet fully elucidated. Therefore, there is an urgent need for comprehensive research focused on the activity of ferroptosis regulators and their functions across different cell types in bone tissue under disease conditions to improve the prospects for therapeutic interventions.

In this review, we have examined how certain pathological conditions, including diabetes, glucocorticoid administration, and postmenopause, contribute to bone-related diseases through the dysregulation of ferroptosis. However, other systemic conditions like aging and hypertension are also linked to the development of bone-related diseases. It remains unclear whether these conditions influence bone health through ferroptosis-dependent mechanisms. Addressing this question could provide valuable insights into the progression of age-related diseases such as OP and OA.

Research has indicated that some ferroptosis inhibitors and activators hold promise as potential therapeutic approaches for bone-related diseases, including certain components of traditional Chinese medicine. However, their therapeutic effects have so far been limited to animal models or *in vitro* cell cultures, with no clinical trials conducted to date. Additionally, the off-target effects and potential side effects of these agents remain not fully understood. Thus, in-depth research into the core mechanisms of ferroptosis inhibitors and activators is needed, along with more clinical trials to support the development of precision treatments for bone-related diseases.

The continuous advancement of innovative technologies and scientific discoveries augurs a future where ferroptosis and its intricate connections to these bone-related diseases are more comprehensively elucidated. This expanded understanding holds the promise of fostering substantial progress in the development of more efficacious treatments for bone-related diseases.

## ABBREVIATIONS

| | |
|---|---|
| **2ME2** | 2-methoxyestradiol |
| **4-HNE** | 4-hydroxynonenal |
| **ACSL4** | Catalyzation of acetyl-CoA synthetase long-chain family member 4 |
| **AGEs** | Advanced glycation end products |
| **BAPTA-AM** | The calcium chelator BAPTA acetoxymethyl ester |
| **BH4** | Tetrahydrobiopterin |
| **CoA** | Coenzyme A |
| **CoQ10** | Coenzyme Q10 |

| | |
|---|---|
| **CoQ10H2** | Reduced coenzyme Q10 |
| **DFO** | Deferoxamine |
| **DMT1** | Divalent metal transporter 1 |
| **DOP** | Diabetic osteoporosis |
| **EC-Exos** | Endothelial cell-secreted exosomes |
| **EPC-EVs** | Extracellular vesicles derived from bone marrow endothelial progenitors |
| **FN** | Ferritin |
| **FSP1** | Ferroptosis suppressor protein 1 |
| **FtMt** | Mitochondrial ferritin |
| **GCH1** | GTP cyclohydrolase-1 |
| **GIOP** | Glucocorticoid-Induced Osteoporosis |
| **GPX4** | Glutathione peroxidase 4 |
| **GSH** | Glutathione |
| **GTP** | Guanosine triphosphate |
| **HIF-1$\alpha$** | Hypoxia-inducible-factor-1-alpha |
| **HSPA5** | Heat shock protein family A family member 5 |
| **IGF-1** | Insulin-like growth factor 1 |
| **IL-1$\beta$** | Interleukin-1 beta |
| **LIP** | Labile iron pool |
| **lipid-ROS** | Lipid-derived reactive oxygen species |
| **LPCAT3** | Lysophosphatidylcholine acyltransferase 3 |
| **MDA** | Malondialdehyde |
| **Nrf2** | Nuclear erythrocyte factor 2 |
| **OA** | Osteoarthritis |
| **OP** | Osteoporosis |
| **OPG** | Osteoprotegerin |
| **OS** | Osteosarcoma |
| **PEITC** | Phenethyl isothiocyanate |
| **PMOP** | Postmenopausal osteoporosis |
| **PUFAs** | Polyunsaturated fatty acids |
| **RCD** | Regulated cell death |
| **ROS** | Reactive oxygen species |
| **SLC7A11** | Solute carrier family 7 member 11 |
| **SLC3A2** | Solute carrier family 3 member 2 |
| **STAT3** | Transcription 3 |
| **STEAP3** | Six-transmembrane epithelial antigen of prostate 3 |
| **T2DOP** | Type 2 diabetes mellitus |
| **t-BHP** | Tert-butyl hydroperoxide |
| **Tf** | Transferrin |
| **TfR1** | Transferrin receptor 1 |
| **UA** | Ursolic acid |

### Funding

This study was supported by the Shandong Provincial Natural Science Foundation of China (Grant Nos. ZR2024MH112, ZR2016HM60 and ZR2024QH628), the Weifang Municipal Health Commission Scientific Research Project (Grant No. WFWSJK-2022-231), the Support Program for Youth Innovation Technology in Colleges and Universities of Shandong Province of China (Grant No. 2019KJK004), the Shandong Medical and Health Science and Technology Development Plan Project (Grant No. 2019WS606) and the Science and Technology Innovation Foundation for the University Students of Shandong Second Medical University (Grant No.2023065). The funders had no role in study design, data collection and analysis, decision to publish, or preparation of the manuscript.

### Grant Disclosures

The following grant information was disclosed by the authors:
Shandong Provincial Natural Science Foundation of China: ZR2024MH112, ZR2016HM60 and ZR2024QH628.
Weifang Municipal Health Commission Scientific Research Project: WFWSJK-2022-231.
Youth Innovation Technology in Colleges and Universities of Shandong Province of China: 2019KJK004.
Shandong Medical and Health Science and Technology Development Plan Project: 2019WS606.
Technology Innovation Foundation for the University Students of Shandong Second Medical University: 2023065.

### Competing Interests

The authors declare that they have no competing interests.

### Author Contributions

- Zihao Wang conceived and designed the experiments, prepared figures and/or tables, and approved the final draft.
- Qiupeng Yan conceived and designed the experiments, prepared figures and/or tables, and approved the final draft.
- Zhen Wang performed the experiments, authored or reviewed drafts of the article, and approved the final draft.
- Zunguo Hu performed the experiments, authored or reviewed drafts of the article, and approved the final draft.
- Chenchen Wang performed the experiments, authored or reviewed drafts of the article, and approved the final draft.
- Xue Zhang performed the experiments, authored or reviewed drafts of the article, and approved the final draft.

- Xueshuai Gao analyzed the data, authored or reviewed drafts of the article, and approved the final draft.
- Xue Bai analyzed the data, authored or reviewed drafts of the article, and approved the final draft.
- Xiaosu Chen analyzed the data, authored or reviewed drafts of the article, and approved the final draft.
- Lingyun Zhang analyzed the data, authored or reviewed drafts of the article, and approved the final draft.
- Danyue Lv analyzed the data, authored or reviewed drafts of the article, and approved the final draft.
- Huancai Liu conceived and designed the experiments, prepared figures and/or tables, and approved the final draft.
- Yanchun Chen conceived and designed the experiments, prepared figures and/or tables, and approved the final draft.

### Data Availability
  This is a literature review.

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
