# Peer review of "Ferroptosis and its implications in bone-related diseases"

_PeerJ, doi:10.7717/peerj.18626_

## Round 0.1 · original submission · Major Revisions

Please address concerns of both reviewers and amend your manuscript accordingly.

Reviewer 1 ·

Basic reporting

- Is the review of broad and cross-disciplinary interest and within the scope of the journal?
The review on ferroptosis in bone-related diseases such as osteoporosis (OP), osteoarthritis (OA), and osteosarcoma (OS) is of broad and cross-disciplinary interest. It is relevant to multiple fields, including:
-Cell Biology and Molecular Biology: The mechanistic discussion of ferroptosis as a form of regulated cell death (RCD) will interest researchers focused on cell death pathways and molecular signaling.
-Orthopedics and Bone Biology: The exploration of how ferroptosis intersects with bone diseases will appeal to researchers and clinicians studying bone physiology, pathology, and treatment strategies.
-Oncology: The review is also relevant to cancer research, particularly in understanding the role of ferroptosis in osteosarcoma, a primary bone cancer.
-Pharmacology and Drug Discovery: The manuscript discusses potential therapeutic targets and drug candidates, which is of interest to researchers involved in drug development and repurposing.
Given the cross-disciplinary nature of ferroptosis research and its application in different bone-related diseases, this review is well within the scope of a journal that focuses on cell biology, molecular medicine, orthopedics, or cancer research.


- Has the Field Been Reviewed Recently?
Yes, similar review for Ferroptosis include:
https://www.nature.com/articles/s41580-020-00324-8
https://www.nature.com/articles/s41580-024-00703-5
https://www.cell.com/cell/pdf/S0092-8674(22)00708-5.pdf

Novel Integration: many of the content described in this manuscript has been well-summarized in above review papers. The novelty of the current manuscript lies in its specific focus on bone-related diseases (OP, OA, OS) and the integration of ferroptosis mechanisms in the context of these diseases, which is not the primary focus of the other reviews mentioned.

Targeted Audience: This review specifically caters to researchers and clinicians in orthopedics and bone biology who may not be deeply familiar with ferroptosis. It provides a tailored overview of the role of ferroptosis in bone diseases, which could help bridge the gap between molecular cell biology and clinical research.

Therapeutic Perspective: The review goes beyond mechanistic studies to discuss potential therapeutic strategies targeting ferroptosis in these diseases, providing practical insights for drug discovery and translational research.

Thus, even if the field has been reviewed recently, this manuscript provides a unique, disease-centered approach to ferroptosis that is likely to be of interest to a different audience and fulfill a niche within the literature.


- Does the Introduction adequately introduce the subject and make it clear who the audience is/what the motivation is?
The Introduction of the manuscript effectively introduces the topic of ferroptosis and its relevance to bone-related diseases. It provides a clear background on ferroptosis as a newly recognized form of regulated cell death (RCD) and its importance in conditions characterized by iron-dependent lipid peroxidation.

Experimental design

-Is the Survey Methodology consistent with a comprehensive, unbiased coverage of the subject? If not, what is missing?
The manuscript briefly mentions the survey methodology in a dedicated section (lines 91–96), stating that articles were retrieved from PubMed, Google Scholar, and Web of Science on the influence of ferroptosis in bone-related diseases from November 1995 to December 2023. Research articles, meta-analyses, editorials, and review articles in English were included, resulting in the selection of 137 articles. However, the description of the methodology is somewhat vague and lacks sufficient detail to ensure that the coverage is both comprehensive and unbiased.


- Are sources adequately cited? Quoted or paraphrased as appropriate?
The manuscript cites numerous sources (roughly 125 citations) to support its claims, referencing both seminal papers and recent studies. Most citations appear to be appropriately paraphrased, and there are no apparent instances of direct quotations without proper attribution



- Is the review organized logically into coherent paragraphs/subsections?
The review is generally well-organized, with clear sections and subsections that guide the reader through the discussion of ferroptosis and its relevance to bone-related diseases. The main sections cover the features and regulators of ferroptosis, major pathways, and the specific links between ferroptosis and OP, OA, and OS. However, there are some areas where the organization could be improved for better clarity and logical flow.

While the sections are logically structured, some subsections could benefit from better internal coherence. For example, the discussion of "Research on GPX4 in ferroptosis" (lines 136–151) jumps between different studies without a clear narrative connecting them. The authors could improve the flow by grouping related studies thematically (e.g., studies on GPX4 regulation vs. studies on GPX4 inhibition) and providing transitional sentences that explain how each study builds on or contrasts with the others.

There is some redundancy, particularly in sections discussing similar pathways across different diseases. For example, the roles of GPX4 and iron metabolism are discussed in multiple disease contexts. While this repetition is partly necessary, the authors could streamline the discussion by introducing a general overview of these pathways first and then focusing on disease-specific nuances.

Validity of the findings

- Is there a well developed and supported argument that meets the goals set out in the Introduction?
The Introduction of the manuscript clearly sets out the goals: to provide a comprehensive overview of ferroptosis and its involvement in bone-related diseases (osteoporosis [OP], osteoarthritis [OA], and osteosarcoma [OS]), and to summarize potential therapeutic strategies targeting ferroptosis for these diseases. To evaluate whether the manuscript meets these goals, we need to assess the development and support of the argument throughout the review.


- Does the Conclusion identify unresolved questions / gaps / future directions?
The Conclusion of the manuscript does provide a summary of the current state of knowledge regarding ferroptosis in bone-related diseases. However, it falls short in identifying specific unresolved questions, gaps in the current understanding, or detailed future directions for research. This is an essential component of a review that aims to guide further research in the field.

The Conclusion lacks specificity in identifying particular unresolved questions that need to be addressed. For example, it does not outline specific molecular mechanisms that remain unclear, potential off-target effects of ferroptosis inhibitors in bone cells, or how differences in bone microenvironments might affect ferroptosis pathways.

The future directions presented are vague and do not provide clear guidance for researchers. The Conclusion should specify key areas where further research is needed, such as:
Understanding the role of specific ferroptosis regulators (e.g., GPX4, SLC7A11) in different types of bone cells (osteoblasts, osteoclasts, chondrocytes).
Investigating how systemic conditions like diabetes or aging affect ferroptosis in bone tissues.
Exploring the development of more selective ferroptosis inhibitors or activators that can target specific pathways without affecting others.

Reviewer 2 ·

Basic reporting

In the manuscript titled “Ferroptosis and its implications in bone-related diseases”- Wang et al. have described the role of ferroptosis in causing OA, OS, and OP. In general, the authors must be commended for diligently including all relevant details and descriptive figures. I have some additional concerns, which are mentioned below:
The manuscript is missing references in lines 65-67.
In the introduction, it is unclear how Ferroptosis and bone-related diseases are linked. The authors should include more details about the link.
In the introduction, the authors should include additional information about the cause of Ferroptosis, symptoms, and treatment options available for the cell death process.
Fig 1. Legend would benefit from a brief description of the pathway.
The authors should add a figure describing the effect of iron metabolism disruption on ferroptosis and the consequent onset of OP.
Additional description is necessary as to how iron metabolism is related to PMOP and how iron starvation affects osteoclasts, along with a figure relevant to the description. In general, a figure for section 3.1.3 is necessary.
Similarly, the manuscript requires a figure relevant to sections 3.2 and 3.3 describing the role of ferroptosis in OA and OS. Fig. 2 does not adequately describe all the events mentioned in the section.

Experimental design

No comments

Validity of the findings

No comments

---

## Round 0.2 · accepted · Accept

All concerns of the reviewers were adequately addressed and revised manuscript is acceptable now.

Reviewer 1 ·

Basic reporting

NA

Experimental design

NA

Validity of the findings

NA

Reviewer 2 ·

Basic reporting

The authors have comprehensively addressed all my concerns and added new figures that adds more value to the manuscript. I have no further issues.

Experimental design

No comment

Validity of the findings

No comment